# Violent deaths following disasters: A retrospective analysis

**Sarah Elizabeth Scales** [1]*, **Lauren C. Camphausen**[2], **Jennifer A. Horney**[2], **Kristina W. Kintziger**[3]

**1** Department of Epidemiology, University of Nebraska Medical Center, Omaha, Nebraska, United States of America, **2** Epidemiology Program, College of Health Sciences, University of Delaware, Newark, Delaware, United States of America, **3** Department of Environmental, Agricultural, and Occupational Health, University of Nebraska Medical Center, Omaha, Nebraska, United States of America

* sscales@unmc.edu

## Abstract

### Introduction

Negative mental health outcomes associated with disaster exposure can increase risks of interpersonal and self-directed violence. However, the link between disaster exposure and increased incidence in violent deaths is not clearly established.

### Study objective

The objective of this work is to assess the incidence of violent deaths in non-disaster and disaster periods across five distinct disaster events.

### Methods

This study uses an ecological, quasi-experimental design to assess violent deaths in pre-disaster and disaster periods across five U.S. states (e.g., North Carolina, Oregon, Oklahoma, Wisconsin, and Colorado) with federally declared disasters resulting from natural hazards (e.g., flood, wildfire, tropical cyclone, storms). Deaths recorded in the National Violent Death Reporting System were used to describe violent deaths with injuries occurring three-months prior to and after disaster onset. Poisson regression with population offsets and fixed effects was used to calculate incidence rate ratios for disaster affected and unaffected counties within the same state, comparing violent death rates in disaster periods with non-disaster periods.

### Results

Most of all deaths were White (80.75%), male (76.87%), and unmarried (65.03%); the median age of decedents was 41.5 years (IQR: 28–55). Overall, the incidence of violent deaths was consistent between disaster and non-disaster periods for both affected (IRR: 0.985; 95% CI: 0.760–1.276) and unaffected counties (IRR: 1.062;

**Data availability statement:** Data cannot be shared publicly because of restrictions imposed for the use of NVDRS Restricted Access Database by the Centers for Disease Control and Prevention. Basic NVDRS data are available through the Web-Based Injury Statistics Query System (https://wisqars.cdc.gov/), and access to the NVDRS Restricted Access Database may be granted from CDC via research application, as detailed here (https://www.cdc.gov/nvdrs/about/nvdrs-data-access.html). These data are administrated by the National Center for Injury Prevention and Control at the Centers for Disease Control and Prevention.

**Funding:** National Violent Death Reporting System (NVDRS) New Investigator Award from the American Public Health Association. The funders had no role in study design, data collection and analysis, decision to publish, or preparation of the manuscript.

**Competing interests:** The authors have declared that no competing interests exist.

95% CI: 0.975–1.158). Rate ratios were heterogenous but not significant across individual disasters, with only non-suicide and all-cause violent deaths increasing significantly following severe storms and flooding in Wisconsin for counties ineligible for public assistance

## Conclusion

The results of this study are consistent with the heterogenous findings on violent deaths and disasters throughout literature. As disasters become more frequent and severe, it is important to further consider the relationship between disaster impacts and negative mental health outcomes, including violent deaths.

## Introduction

Disasters, by definition, create circumstances that are beyond the standard coping capacity of individuals or communities to respond [1]. These disaster effects are not limited to physical or quantifiable impacts – such as destruction of property, damages to civilian and vital infrastructure [2], injuries and fatalities – but also include indirect and otherwise 'invisible' impacts. It is widely accepted that disasters have severe impacts on the mental health and wellbeing of affected populations, and poor mental health can increase the risk of interpersonal and self-directed violence. However, the relationship between disasters and violence is not well understood.

### Mental health and disasters

The mental health impacts of disasters are fluid, with the likelihood of negative mental health outcomes fluctuating depending on the severity, scale, and exposure time of the event. Palinkas and Wong (2020) outlined the challenges with documenting and effectively addressing the disparate negative mental health impacts of acute, subacute, long-term, and existential impacts of disasters [3]. Berry et al. (2010) developed a causal framework for understanding the mental health impacts of climate-related disasters based on a literature review [4]. Disasters affect communities and individual physical and mental health both directly and indirectly. Injuries and fatalities – from heat illness, crush injuries, drowning, shock, among others – are examples of direct effects on physical health [5,6]. Further, direct injuries resulting from a given disaster can also contribute to sustained negative mental health outcomes such as post-traumatic stress, depression, and anxiety [7].

In addition to immediate, direct health impacts, disaster-related factors can indirectly exacerbate negative mental and physical health. For example, income loss, especially for those with lower socioeconomic status, can cause stress, anxiety, and depression [8,9]. Increasing economic inequality, driven in part by more frequent and severe disasters resulting from climate-related hazards can further exacerbate the negative mental health impacts these events [10]. Community impacts can mediate both individual mental and physical health outcomes, while physical and mental

health are interdependent [4], although somatization of disaster-related trauma is not universally observed across disaster types [11,12].

A range of hazard types can lead to poor mental health outcomes, and these outcomes can occur both acutely and longitudinally. There is not a standard timeframe for assessing mental health impacts of disasters resulting from natural hazards. Most studies assessing mental health following disasters are cross-sectional or repeated cross-sectional studies, with fewer studies assessing impacts longitudinally, and the timepoints considered vary across studies [7]. Across the disaster and recovery cycle, post-traumatic stress, anxiety, and depression increase in populations experiencing wildfires and exposed to wildfire smoke [13,14]. Pietrzak et al., (2012), assessed mental health disorders following Hurricane Ike in Galveston and Chambers counties, Texas, at four time points, and found that post-traumatic stress was most prevalent between 2- and 5-months following the event [15]. Further, the prevalence of all mental health conditions attributed to Hurricane Ike decreased over time. In winter storm Uri, Texans with disabilities, racial and ethnic minorities, and those with prior traumatic storm experiences were more likely to have anxiety or depression six months after the disaster event [16]. Following the 2016 Fort McMurray wildfire in Alberta, Canada, 15% of displaced individuals reported depressive symptoms a year after the wildfire [17]. Sri Lankans from coastal districts who experienced the Indian Ocean tsunami of 2004 were more likely to report partial or post-traumatic stress disorder eight years after the event compared to those without exposure [18].

### Interpersonal and self-directed violence and disasters

One of the key considerations for the relationship between disasters and interpersonal violence is the mediating role of external stressors in instigating new or exacerbating existing poor mental health. Mental illness is not, in and of itself, a predictor of violent behaviors; however, in conjunction with other circumstantial factors, such as substance use, disaster exposure, or other external stressors, poor mental health can contribute to violent behaviors [19]. While the relationship between natural hazard exposure and death by homicide or suicide is not well established, it is clear that natural hazard exposure increases perpetration of intimate partner violence and family violence [20] and extreme heat exposure increases a the incidence of various violent behaviors [21]. There are a number of potential confounding and moderating variables for the relationship between disaster exposure and violent deaths, but measurement of these variables is particularly challenging [22]. Matsubayashi et al. (2013) found disparate effects of disasters on suicide rates at the prefecture level in Japan [23]. Rates were dependent on the severity of damage – measured physical and economic losses – resulting from the hazard, and rates varied across age and sex combinations. While findings on the relationship between, and temporality of, violent deaths after disaster exposure are not consistent, negative mental health impacts – including post-traumatic stress, depression, and anxiety – which can contribute to violent behaviors are consistently observed throughout the literature.

Given these complexities, it is critically important to consider the mental health impacts of disasters, including not only violent behaviors but also violent deaths. The aims of this work are 1) to characterize violent deaths in non-disaster and disaster periods and 2) to assess changes in incidence of violent deaths for counties eligible and ineligible for public assistance in non-disaster and disaster periods. By assessing deaths resulting both from suicides and interpersonal violence, this study adds to the literature base for understanding sequelae of violence in disaster contexts.

## Materials and methods

### Study description and population

The National Violent Death Reporting System records homicide and suicide deaths across the United States. Five states were selected for this study – North Carolina, Oregon, Oklahoma, Wisconsin, and Colorado. These states were selected based on 1) the availability of data for the three months before and after a federally declared disaster and 2) the coverage

of priority hazard types represented by the instigating hazard events (i.e., flooding, storms, tropical cyclone, fire), and 3) the absence of co-occurring, compounding, or cascading disasters.

The three-month disaster period was selected to measure acute disaster-related mental health impacts rather than more longitudinal impacts outside of the immediate response period. FEMA public assistance eligibility was used as a proxy measure for disaster exposure because federal assistance eligibility requires a community to have loss and damage that surpasses local and state-level capacity to address resulting acute needs, response, and recovery. It is important to note, however, that smaller scale disasters that are not granted federal assistance can still have tremendous effects on impacted communities, and, even within those eligible for public assistance, there is appreciable variation in need. States and associated hazard information are summarized in **Table 1**.

## Ethics statement

This work did not include human subjects research; data were de-identified with no way of ascertaining the identities or individuals represented in the data set. Therefore, review from an institutional review board was not necessary or required. Secondary, deidentified death record data were used in this study, and oral or written consent could not be given and was not required. Data access was granted March 17, 2024. All findings are reported in compliance with suppression guidelines for ensuring anonymity from the Centers for Disease Control and Prevention.

## Measures

Deaths from each state were aggregated at the county level, and deaths were assigned to the individual's county of residence. The pre-disaster period was defined as the period three months directly preceding the recorded disaster start date in the corresponding federal disaster declaration (**Fig 1**). The disaster period covered the three months – operationalized as calendar months (see date ranges in Table 1)– following the disaster start date, as defined in the federal disaster declaration. The first date of the disaster period is the date of event onset. This time frame was selected to capture the impacts of acute stress from given disaster events and response as opposed to the longer term stressors related to recovery and rebuilding. Because NVDRS only records deaths, and to ensure that injuries leading to death were accurately captured in the pre-disaster or disaster period, injury date (rather than death date) was used to assign the individual to the period. Using injury date ensures that the causative injury event preceding a violent death occurred in the termporal range of interest. To assess the potential influence of disaster exposure on violent deaths, data were stratified by eligibility for federal disaster public assistance, as defined by federal disaster declarations [24]. Under the Stafford Act, state governors

**Table 1. States and associated hazard information.**

| State | FEMA Region | Disaster Declaration(s) | Date Ranges Pre-Disaster Disaster | Hazard/ Hazard Type | Counties Eligible for Public Assistance | Total Counties |
|---|---|---|---|---|---|---|
| Colorado | VIII | DR-4143-CO DR-4134-CO | 11 Mar – 10 Jun 2013 11 Jun – 10 Sep 2013 | Royal Gorge and Black Forest wildfires | 2 | 64 |
| North Carolina | IV | DR-1608-NC | 11 Jun– 10 Sep 2005 11 Sep – 10 Dec 2005 | Hurricane Ophelia | 10 | 100 |
| Oklahoma | VI | DR-4299-OK | 13 Oct 2017–12 Jan 2018 13 Jan – 12 Apr 2018 | Severe winter storm | 11 | 77 |
| Oregon | X | DR-1733-OR | 1 Aug – 30 Nov 2007 1 Dec 2007–30 Mar 2008 | Storms, flooding, and mass movement | 9 | 36 |
| Wisconsin | V | DR-4141-WI | 20 Mar – 19 Jun 2012 20 Jun – 19 Sep 2012 | Severe storms, mudslides, and floods | 8 | 72 |

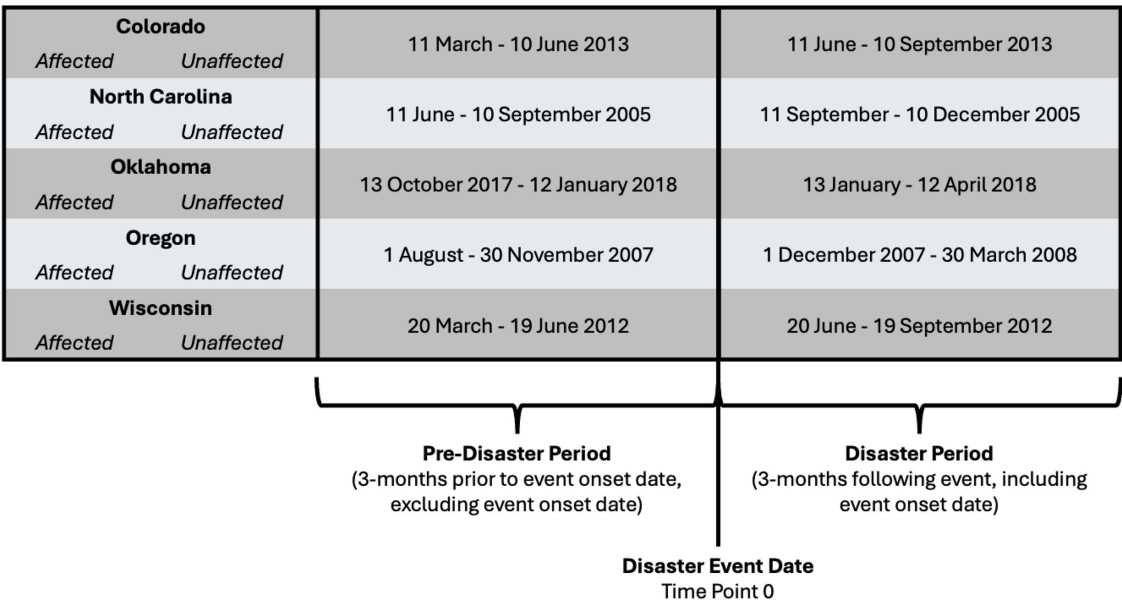

**Fig 1. Schematic showing pre-disaster and disaster periods, normalized to a common timepoint.**

and tribal executives can request federal aid to address materiel, recovery, and economic needs, among others, that surpass local, state, or tribal capacity to address these needs [25]. Upon an event being declared a major disaster or emergency by the President of the United States, public assistance grants are disseminated through the U.S. Federal Emergency Management Agency (FEMA) to aid in community and public agency response and recovery efforts [26]. Disaster declarations specify which counties within a state are eligible for public, individual, or other forms of assistance. For this study, public assistance eligibility was used as a proxy for disaster-affected counties: counties eligible for public assistance were 'affected' while counties in the same state that were ineligible for public assistance were 'unaffected.'

Sociodemographic variables were used to describe deaths. To account for small cell counts and suppression, race (White only/ other), sex (male/ female), and marital status (married or partnered/other) were included as binary variables. Education was measured as a categorical variable (high school or less/ post-high school education/ missing). Age was treated as a continuous variable. Because the unit of analysis for this study was the aggregate of affected and unaffected counties, sociodemographic characteristics are reported to contextualize the underlying population but are not included in regression models.

## Analysis

Individual-level frequencies and proportions representing the underlying demographics of deaths recorded in the NVDRS were calculated. To compare deaths among those eligible for public assistance and those not eligible in pre-disaster and disaster periods, chi-square p-values are reported for categorical variables (i.e., race, sex, marital status, and education), and Kruskal-Wallis chi-square p-values are reported for median age and interquartile range. Crude incident rates were calculated with 95% confidence intervals (S1–S3 Tables).

## Model specification for incidence rate ratios

This study uses an ecological pre-post (i.e., pre-disaster and disaster period) design with within-state internal controls (i.e., affected an unaffected counties). Incidence rate ratios comparing violent deaths in disaster periods to pre-disaster

periods were calculated using Poisson regression with a population denominator offset and log link, using the GENMOD procedure in SAS v9.4 (Cary, NC). Overdispersion was assessed by comparing the mean and variance of the count data, and Poisson regression was an appropriate modelling selection. Incident rate ratios and 95% confidence intervals are reported for all states together, as well as stratified results for each state individually.

## Unit of analysis

Counties were aggregated into those ineligible and those eligible for public assistance, and results are stratified by these units of analysis.

## Predictor variables

For all models, disaster period (yes/no) was treated as a fixed effect. When controlling for state to generate overall rate ratios comparing non-disaster and disaster periods, residence state was treated as a fixed effect.

## Outcomes

Models were specified with all deaths, suicide deaths only, and other violent deaths only, as outcomes of interest. ICD-10 codes corresponding to intentional self-harm or sequelae of intentional self-harm were used to determine suicide deaths. All other deaths were considered 'non-suicide violent deaths.' All violent deaths include any death recorded in the NVDRS database, including both suicide and other forms of violent deaths. Other violent deaths include homicides, deaths in individuals killed by law enforcement officers in the line of duty, violent deaths of undetermined intent (including manslaughter), and unintentional firearm deaths.

## Results

There were a total of 2322 deaths recorded in the study period for all included states. In the pre-disaster period, there were 1151 deaths (49.57%), compared to 1171 deaths in the disaster period (50.43%). Most deaths occurred in counties not eligible for public assistance (n = 2091, 90%). Suicides (n = 1685) accounted for 72.57% of all violent deaths. For deaths that occurred in the disaster period, 12.84% of causative injuries (n = 14) occurred in a county other than the individual's county of residence; 11 of which occurred in counties not eligible for public assistance. Half of counties eligible for public assistance were classified as urban and half as mostly or completely rural in the 2010 census. Death counts for the entire study period, in the pre-disaster and disaster periods, and by public assistance eligibility are summarized in **Table 2**. Stratification by both disaster period and public assistance is not shown due to suppressed cell counts.

Table 2. Death frequency for pre-disaster and disaster periods; eligibility for public assistance.

| | Overall | | Pre-Disaster | | Disaster | | Ineligible for Public Assistance | | Eligible for Public Assistance | |
|---|---|---|---|---|---|---|---|---|---|---|
| | n | % | n | % | n | % | n | % | n | % |
| Colorado | 482 | 20.76 | 223 | 19.37 | 259 | 22.12 | 399 | 19.09 | 83 | 35.78 |
| North Carolina | 788 | 33.94 | 416 | 36.14 | 372 | 31.77 | 725 | 34.69 | 63 | 27.16 |
| Oklahoma | 344 | 14.81 | 167 | 14.51 | 177 | 15.12 | 335 | 16.03 | 9 | 3.88 |
| Oregon | 271 | 11.67 | 149 | 12.95 | 122 | 10.42 | 217 | 10.38 | 54 | 23.28 |
| Wisconsin | 437 | 18.82 | 196 | 17.03 | 241 | 20.58 | 414 | 19.81 | 23 | 9.91 |
| All States | 2322 | 100 | 1151 | 100 | 1171 | 100 | 2091 | 100 | 231 | 100 |

The distribution of sociodemographic characteristics of individual deaths are presented in **Table 3**. The median age of all deaths was 41.5 years (28–55 years). Most of all deaths were white (80.75%) and male (76.87%). Roughly 35% of decedents were married, in a civil union, or partnered. Educational attainment was not reported for 54% of deaths. Race, marital status, and education were statistically different between deaths with initial injury occurring in counties receiving public assistance compared to those not, while median age and sex were similar between the two.

Incidence rate ratios comparing violent deaths with initial injury during the disaster period to those with initial injury prior to the disaster period for counties eligible and ineligible for public assistance are shown in **Table 4**. Overall, the incidence of violent deaths was consistent between disaster and non-disaster periods for both affected (IRR: 0.985; 95% CI: 0.760–1.267) and unaffected counties (IRR: 1.062; 95% CI: 0.975–1.158). The overall suicide mortality rate did not change significantly between non-disaster and disaster periods for either affected (IRR: 0.891; 95% CI: 0.670–1.184) or unaffected counties (IRR: 1.045; 95% CI: 0.944–1.156). Non-suicide violent deaths did not change significantly between periods for either affected (IRR: 1.647; 95% CI: 0.857–3.166) or unaffected counties (IRR: 1.108; 95% CI: 0.944–1.301). Only Wisconsin saw a significant increase in non-suicide violent deaths for unaffected counties (IRR: 2.522; 95% CI: 1.600–3.979), which drove the observed increased rate of all violent deaths in the disaster period for unaffected counties in Wisconsin.

**Table 3. Descriptive statistics of sociodemographic characteristics of all deaths.**

| | | | Pre-Disaster | | | Disaster | | |
|---|---|---|---|---|---|---|---|---|
| | | Overall | No Public Assistance | Public Assistance | Chi-square p-value | No Public Assistance | Public Assistance | Chi-square p-value |
| | | n (%) | n (%) | n (%) | | n (%) | n (%) | |
| Race | White only | 1875 (80.75) | 831 (80.76) | 109 (89.34) | 0.021 | 841 (79.26) | 94 (84.45) | 0.123 |
| | Other | 447 (19.25) | 198 (19.24) | 13 (10.66) | | 220 (20.74) | 16 (14.55) | |
| Sex | Male | 1785 (76.87) | 780 (75.80) | 98 (80.33) | 0.266 | 822 (77.47) | 85 (77.27) | 0.962 |
| | Female | 537 (23.13) | 249 (24.20) | 24 (19.67) | | 239 (22.53) | 25 (22.73) | |
| Marital Status | Married or partnered | 812 (34.97) | 338 (32.85) | 51 (41.80) | 0.048 | 683 (64.37) | 65 (59.09) | 0.272 |
| | Other | 1510 (65.03) | 691 (67.15) | 71 (58.20) | | 378 (35.63) | 45 (40.91) | |
| Education | High school or less | 701 (30.19) | 321 (31.20) | 24 (19.67) | 0.015 | 560 (52.78) | 65 (59.09) | 0.347 |
| | Post-high school education | 367 (15.81) | 160 (15.55) | 17 (13.93) | | 329 (31.01) | 27 (24.55) | |
| | Missing | 1254 (54.01) | 548 (53.26) | 81 (66.39) | | 172 (16.21) | 18 (16.36) | |
| Age | **Median (IQR)** | 41.5 (28-55) | 44 (28-54) | 40.0 (27-54) | 0.521* | 42 (28-55) | 42 (23-60) | 0.882* |

\* Indicates a Kruskal-Wallis chi-square p-value. All other p-values were from a standard (Pearson's) chi-square test.

**Table 4. Incidence rate ratios comparing disaster period violent death, suicide, and non-suicide violent death rates with prior period violent death, suicide, and non-suicide violent death rates.**

| | All Violent Deaths | | | | Suicide Deaths | | | | Non-Suicide Violent Deaths | | | |
|---|---|---|---|---|---|---|---|---|---|---|---|---|
| State | No PA | | Public Assistance | | No PA | | Public Assistance | | No PA | | Public Assistance | |
| | IRR | 95% CI | IRR | 95% CI | IRR | 95% CI | IRR | 95% CI | IRR | 95% CI | IRR | 95% CI |
| CO | 1.159 | 0.952-1.412 | 0.930 | 0.605-1.431 | 1.157 | 0.935-1.432 | 0.784 | 0.482-1.274 | 1.173 | 0.699-1.968 | 1.833 | 0.678-4.957 |
| NC | 0.928 | 0.802-1.074 | 0.887 | 0.535-1.469 | 0.903 | 0.743-1.097 | 0.826 | 0.467-1.462 | 0.962 | 0.773-1.198 | 1.155 | 0.388-3.438 |
| OK | 1.125 | 0.908-1.394 | 1.000 | 0.250-3.998 | 1.249 | 0.970-1.609 | 1.000 | 0.250-3.998 | 0.857 | 0.570-1.290 | x | x |
| OR | 0.962 | 0.736-1.259 | 1.126 | 0.660-1.920 | 0.951 | 0.707-1.278 | 1.176 | 0.679-2.038 | 1.019 | 0.538-1.932 | 0.523 | 0.047-5.765 |
| WI | 1.244 | 1.025-1.510 | 1.167 | 0.505-2.695 | 1.037 | 0.834-1.290 | 0.750 | 0.298-1.889 | 2.522 | 1.600-3.979 | x | x |
| All | 1.062 | 0.975-1.158 | 0.985 | 0.760-1.276 | 1.045 | 0.944-1.156 | 0.891 | 0.670-1.184 | 1.108 | 0.944-1.301 | 1.647 | 0.857-3.166 |

## Discussion

Zahran et al., (2009), described the relationship between natural hazards and social order, including violent crimes, and found an increased count of domestic violence crimes [27]. However, this does not naturally extend to violent deaths. While it is widely acknowledged that exacerbated poor mental health and violent behaviors can precipitate an increase in violent deaths, the literature on violent deaths in disaster contexts centers on suicide deaths rather than also considering homicide and manslaughter. The rate of non-suicide violent deaths increased in counties eligible for public assistance in the state of Colorado in the three months following wildfires, but the overall violent death rate for the state did not change. However, the increased rate of non-suicide violent deaths in Wisconsin counties ineligible for public assistance drove the observed increase in all violent deaths. Investigation into the consistency and causality of disaster-related escalations of increased violent behaviors and resulting deaths is needed to improve evidence-informed harm reduction policy and practice. The suicide mortality rate only increased for Oklahoma counties ineligible for public assistance in the 3-months following a winter storm. Rates were consistent in all other states individually and collectively.

Kolves and Kolves (2013) found discrepancies across 42 articles assessing the effect of disaster exposure on suicide attempts and mortality, with study of results assessing suicide mortality varying inconsistently across hazard type, location, time, or severity of impact [28]. In an analysis of Rhode Island's Violent Death Reporting and Syndromic Surveillance Systems data, Barkley (2023), found that suicide deaths were consistent over time across sex, age, and race [29]. Over the study period – 2012–2021 – Rhode Island was affected by ten billion-dollar disaster events, including tropical cyclones, severe storms, drought, and extreme winter weather [30]. Two years after the Indian Ocean tsunami, suicide rates in affected and unaffected areas of Sri Lanka remained relatively consistent with rates from the two years prior to the disaster.

The scale of economic loss and damage and severity of morbidity and mortality in a disaster can contribute to higher levels of stress among survivors. Of the events included in this study, none surpassed the billion-dollar mark, as assessed by the National Oceanic and Atmospheric Administration. There were no direct deaths resulting from the Royal Gorge wildfire, despite more than 3200 acres burning [31], while there were two confirmed deaths and nearly 14300 acres burned in the Black Forest wildfire [32]. One attributable death – a car accident – was recorded for Hurricane Ophelia (NASA, 2005). However, the economic damage and physical impacts of the five events included in this study, do not completely explain the inconclusive findings. Total financial impacts are only one way to measure the severity of a disaster, and even disasters that affect smaller communities and/or have more attenuated impact when compared to other events severely impact the communities that experience them. In this study, population offsets were used to account for differences in population size and potential discrepancies in scale. While typically considered for the most affected populations, the negative effects of a disaster are not constrained to only communities affected by the event [33]. Communities tangentially affected by, or even removed from, an event can also experience negative mental health outcomes related to seeing or hearing about the scale of economic loss and damage and severity of human impacts; becoming host communities to temporarily or permanently displaced populations; witnessing the toll of the event on loved ones or friends; or providing relief via mutual aid to the more directly affected areas [34–36]. While these considerations are discussed in the context of this study, any unmeasured spillover effects would attenuate measured incidence.

Protective actions such as evacuation and sheltering-in-place are additional factors that should be considered in the relationship between disasters and violent deaths. Both evacuation and shelter-in-place orders – as well as any disaster-related displacement – contribute to stress, exacerbated poor mental health, and new-onset mental illness [8,37]. In our study, the displacement profiles of the individual events are different, with events like Hurricane Ophelia in North Carolina or the Royal Gorge and Black Forest wildfires in Colorado prompting evacuations for some or all of the affected areas. Alternatively, severe winter weather in Oklahoma prompted shelter-in-place orders. Severe storms and floods, such as those occurring in Oregon and Wisconsin, resulted in both shelter-in-place and evacuation orders, depending on the location, timing, and severity of disaster impacts. However, observed incidence rates were not unidirectional across typical

protective actions. Future studies should account for the role of displacement and sheltering in the relationship between disasters, stress, and violent deaths.

Another explanation for the discrepancy between the consistently observed increase in interpersonal violence and negative mental health outcomes immediately following a disaster, compared to the number of associated deaths in the same timeframe, could be that the resources and external supports that are part of immediate recovery efforts are attenuating the escalation of interpersonal and self-directed violence. For example, the Disaster Distress Hotline was started by the Substance Abuse and Mental Health Services Administration in 2012, to offer around the clock support for those affected by disasters caused by natural and other hazards [38]. Behavioral health trainings and toolkits are available to support disaster response and recovery through SAMHSA's Disaster Technical Assistance Center [39]. While these resources are available, many on-the-ground, tangible, and skilled supports, such as those provided through the American Red Cross or locally deployed counseling services, are not widely available outside of the acute phases of recovery.

While trauma related to disaster exposure can dissipate over time, anxiety and depression associated with disaster exposure are often persistent [40] necessitating targeted programming to address not only these direct mental health impacts but also the associated long-term sequelae. As with public health broadly, a significant portion of mental health programming in disaster response is built from experience and subject matter expertise but is then never fully evaluated to contribute to the development of a robust evidence base [41]. Future work should evaluate the utilization and effectiveness of these programs in disaster affected communities.

## Limitations

In addition to the limitations and implications on study results previously discussed, there are other important limitations to consider. Not all states have reported to NVDRS, and the years with reporting coverage are not standard across states. Further, coverage within reporting states is not necessarily inclusive of all counties, with some states only reporting for a subset of their population. In selecting states for this study, we considered all of these factors, as well as the presence of a federally declared disaster within the timeframe of available data.

For the purposes of this study, we were interested in assessing the rate of deaths resulting from both interpersonal and self-directed violence in the acute period following disaster events. The relevance of this timeframe was demonstrated by Pietrzak et al., who found the prevalence post-traumatic stress peaked between 2- and 5-months following the event [15]. Some work suggests that deaths resulting from self-directed violence occur at a longer-term interval, with increased rates 2-years following exposure to a disaster event [28,42]. As reporting on violent deaths improves, temporal aspects should be considered within the larger framework of understanding the effects of disasters on violent deaths, particularly as disaster recovery complexity extends recovery periods and focuses on building resilience.

This study aggregates data at the county level. To reduce potential biases in exposure classification, we assigned deaths to the county of residence, not county of injury or death. For example, a resident of one county could have died in a neighboring county ineligible for public assistance dependent upon the location and accessibility of skilled trauma centers. A more granular understanding of individual-level disaster exposures could provide different or more nuanced findings. Known confounders of the relationship between disaster exposure and violent deaths at the individual-level were not available and/or comparably measured at the county-level for all counties in the study. Future work should not only account for potential aggregate-level confounders but also consider confounders or effect modifiers that differ between suicide and other violent deaths individually.

We used eligibility for federal public assistance to determine if a county was disaster affected. However, the effect of a disaster on a community is considerably more nuanced than the presence or absence of a federal disaster declaration. For example, populations within a community could still experience highly disparate disaster impacts from one another, underscoring the importance of capturing disaster exposure on a more granular scale. Different classifications to better capture the nuance of disaster exposure should be explored in future studies that aim to assess the causative relationship

of disasters on suicide and violent death mortality rates. Additionally, more complete reporting on decedent's disaster exposure within the NVDRS reporting framework could greatly improve the feasibility of assessing the impact of disasters on violent deaths at the individual-level.

## Conclusions

The results of this study are in line with the heterogenous findings assessing the relationship between disaster exposure and violent deaths. Given the limited studies on this topic, as well as the complicated relationship between disaster exposure, mental health, violence, and other external factors, it is important to continue to investigate these relationships to better promote mental health and wellbeing as disasters become more frequent and severe.

## Supporting information

**S1 Table. Rates for all violent deaths in pre-disaster and disaster periods stratified public assistance eligibility.**
(DOCX)

**S2 Table. Rates for suicide deaths in pre-disaster and disaster periods stratified public assistance eligibility.**
(DOCX)

**S3 Table. Rates for other violent deaths in pre-disaster and disaster periods stratified public assistance eligibility.**
(DOCX)

## Author contributions

**Conceptualization:** Sarah Elizabeth Scales, Jennifer A Horney, Kristina W Kintziger.

**Data curation:** Sarah Elizabeth Scales.

**Formal analysis:** Sarah Elizabeth Scales.

**Funding acquisition:** Sarah Elizabeth Scales, Jennifer A Horney.

**Methodology:** Sarah Elizabeth Scales, Jennifer A Horney, Kristina W Kintziger.

**Validation:** Kristina W Kintziger.

**Writing – original draft:** Sarah Elizabeth Scales.

**Writing – review & editing:** Sarah Elizabeth Scales, Lauren C Camphausen, Jennifer A Horney, Kristina W Kintziger.

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
