## [Decision Letter · Decision Letter 0]

11 Aug 2025

Violent Deaths Following Disasters: A Retrospective Analysis

PLOS ONE

Dear Dr. Scales,

Thank you for submitting your manuscript to PLOS ONE. After careful consideration, we feel that it has merit but does not fully meet PLOS ONE’s publication criteria as it currently stands. Therefore, we invite you to submit a revised version of the manuscript that addresses the points raised during the review process.

We look forward to receiving your revised manuscript.

Kind regards,

Carolyn Chisadza

Academic Editor

PLOS ONE

Journal Requirements:

“National Violent Death Reporting System (NVDRS) New Investigator Award from the American Public Health Association”

Please state what role the funders took in the study.  If the funders had no role, please state: 'The funders had no role in study design, data collection and analysis, decision to publish, or preparation of the manuscript.'

Additional Editor Comments:

In particular, the authors should pay close attention to clearly outlining their methodology, discussing their results and the limitations of the study.

Reviewers' comments:

Reviewer's Responses to Questions

**Comments to the Author**

1. Is the manuscript technically sound, and do the data support the conclusions?

Reviewer #1: Yes

Reviewer #2: Partly

2. Has the statistical analysis been performed appropriately and rigorously?

Reviewer #1: I Don't Know

Reviewer #2: No

3. Have the authors made all data underlying the findings in their manuscript fully available?

Reviewer #1: No

Reviewer #2: No

4. Is the manuscript presented in an intelligible fashion and written in standard English?

Reviewer #1: Yes

Reviewer #2: Yes

Reviewer #1: Thank you for the opportunity to review this paper.

This is a neat and to-the-point study showing inconclusive results, but what makes the paper interesting is that the authors tried to address an important research gap in a creative way using existing data. What I feel is that the authors should explore this more in depth and reflect more on the methodological choices and discuss them more thoroughly, as this can pave the way for future studies that may yield more conclusive results.

I have noted some comments aligned with this :

Methodology:

- The study design is not clearly presented. Perhaps adding a study schematic showing how the temporal associaition was assessed would be helpful.

- Suggest to clearly state (with e.g. sub-sections) what is the main outcome of interest, and eventually secondary outcomes of interest . what are problems or advantages of putting all violent deaths together? (to consider for limitations)

- Clarify the exposure and control. 3 months (what threshold was used operationally , 90 days?), before and after a federally declared disaster -- any information on what qualifies as a federal disaster?

- table IV formatting makes it very difficult to read

- IRRs were calculated, but this implies that incidence rates were calculated, but the methods only talk about a population denominator offset . does this mean cases per person-time was not calculated? If calculated please add as appendix.

Discussion:

- would have appreciated more in-depth discussion of the methodological choices of this study and put them in context with other existing studies (? ) assessing a temporal association between disasters and any potential longer-term effects.

- the limitations section is too short and not correctly addressing all potential biases or confounding.

Reviewer #2: This paper provides an empirical analysis examining the impact of natural disasters on violent deaths across five diverse states, each experiencing distinct natural disaster events. Using county-level data, the paper compares violent death counts—including suicides and non-suicide violent deaths—before and after disaster events in both disaster-affected and unaffected counties. The authors adopt a descriptive analytic approach, contrasting patterns of violent deaths pre- and post-disaster across these groups. However, the robustness and clarity of the findings could be significantly enhanced by explicitly employing a rigorous quasi-experimental design, such as a difference-in-differences (DiD) or event-study framework. Baker et al. (2022) and Obradovich et al. (2018) provide valuable examples of how to implement these methodologies effectively in related contexts.

In addition, the discussion of results (e.g., increased incidence of violent deaths post-disaster in affected versus unaffected areas) would benefit from greater consideration of potential spillover effects between counties, a critical factor given the complexity of post-disaster community dynamics.

Moreover, several aspects of the methodological and analytical choices require further clarity and justification—for instance, the rationale for selecting a three-month period as representative of short-term mental health impacts, and the specific inclusion and treatment of age as a continuous covariate in regression analyses. The manuscript would benefit from improved precision and clarity in reporting statistical methods and results. Specifically, authors should clarify the statistical tests used, implement appropriate adjustments for multiple testing, and explicitly define their regression modeling approach. These refinements would substantially strengthen the manuscript's rigor and interpretability.

Major comments:

1. Improve the statistical design of this study by employing a quasi-experimental design, such as difference-in-difference or event-study frameworks.

Here is a helpful background resource:

Baker, A. C., Larcker, D. F., & Wang, C. C. (2022). How much should we trust staggered difference-in-differences estimates? Journal of Financial Economics, 144(2), 370-395.

Here is a study that applies this methodology to look at the impacts of Katrina on mental health in affected and unaffected counties:

Obradovich, N., Migliorini, R., Paulus, M. P., & Rahwan, I. (2018). Empirical evidence of mental health risks posed by climate change. Proceedings of the National Academy of Sciences, 115(43), 10953-10958.

2. Page: 18 lines 220-224: Improve robustness of the discussion of the increased incidence of violent deaths post-disaster in affected and unaffected areas. Please also discuss spillover effects across counties post-disaster.

Minor comments:

Page: 11 lines 76-79: This passage is difficult to follow. Please expand on the community impacts in the context of disasters. It is also unclear what the Australian experience is contradicting to.

Page: 12 line 107: Expand this discussion to extreme weather exposure and violence literature:

Ranson, M., 2014. Crime, weather, and climate change. J. Environ. Econ. Manag. 67, 274–302. https://doi.org/10.1016/j.jeem.2013.11.008

Hsiang, S., Kopp, R., Jina, A., Rising, J., Delgado, M., Mohan, S., Rasmussen, D.J., Muir-Wood, R., Wilson, P., Oppenheimer, M., Larsen, K., and Houser T. (2017). Estimating economic damage from climate change in the United States, Science, 356, 1362–1369

Page: 14 lines 135-137: Please support the specific selection of three months to represent short-term mental health effects. Are the results sensitive to the specific choice of the timeframe?

Page: 15 lines 151-152: Earlier you note that age was treated as a continuous variable. How was it introduced in the Poisson regression of aggregate counts in a county/time period?

Page: 15 lines 154-156: Provide additional information on the regression analysis performed. What was the unit of observation? What controls, beyond affected status and pre/ post time period were included? How the overall statistics were obtained -a pooled regression? a fixed effects regression? A meta-analysis of the regression results across states?

Page: 16 lines 178-181: Please elaborate on what the differences were and also in which period (pre/post) these differences were noted.

Page: 16 lines 193-194: Please eliminate discussion of results that did not show up as significant patterns. If a result is not statistically significant, we cannot make claims about protectiveness. Given these wide confidence intervals, it just as easily found be null or >1.

Page: 26 Table I: Report the total number of counties in the state.

Page: 27 Table II: Add crude incidence rates to this table.

Page: 28 Table III: Improve labeling. Which test is Chi-Square and which test is Kruskal-Wallis? Also, apply controls for the multiple testing to the results in this table.

Page: 29 Table IV: Apply adjustments for multiple testing in this table.

**Do you want your identity to be public for this peer review?** For information about this choice, including consent withdrawal, please see our Privacy Policy

Reviewer #1: **Yes: ** Maria Moitinho de Almeida

Reviewer #2: No

---

## [Author Response · Author response to Decision Letter 1]

21 Aug 2025

We have addressed all formatting and data availability statement concerns raised by the Editor. These changes are manifest in the submitted revision and appropriate web-form fields.

We thank the reviewers for the substantive responses. We have addressed these comments below.

Reviewer #1: Thank you for the opportunity to review this paper. This is a neat and to-the-point study showing inconclusive results, but what makes the paper interesting is that the authors tried to address an important research gap in a creative way using existing data. What I feel is that the authors should explore this more in depth and reflect more on the methodological choices and discuss them more thoroughly, as this can pave the way for future studies that may yield more conclusive results. I have noted some comments aligned with this:

Methodology:

• The study design is not clearly presented. Perhaps adding a study schematic showing how the temporal association was assessed would be helpful.

o Figure 1 has been added to visually show the temporality of 3-months before and 3-months after the initial disaster event date, and additional clarification has been added in-text, reading: “The disaster period covered the three months – operationalized as calendar months – following the disaster start date, as defined in the federal disaster declaration.”

• Suggest to clearly state (with e.g. sub-sections) what is the main outcome of interest, and eventually secondary outcomes of interest.

o Subsections and explicit definitions have been added under the Analysis section of Methods and Materials. The methods section and results have been updated following updated analysis in response to reviewers.

• What are problems or advantages of putting all violent deaths together? (to consider for limitations).

o We have addressed this in two ways: 1) results for all deaths, as well as stratified by violent death category (i.e., suicide versus other violent death), are presented; and 2) added a brief discussion in the discussion/limitations.

• Clarify the exposure and control. 3 months (what threshold was used operationally , 90 days?), before and after a federally declared disaster.

o Figure 1 has been added to visually show the temporality of 3-months before and 3-months after the initial disaster event date. These periods were defined as calendar months before and after the date of disaster onset. Federal disaster declarations can happen in real-time, but they are often issued post-event after significant damage is reported or accrued. Therefore, we define the start of the disaster period as the event date included in the declaration rather than the date of the declaration. The text under Measures now reads: “The disaster period covered the three months – operationalized as calendar months – following the disaster start date, as defined in the federal disaster declaration.”

• Any information on what qualifies as a federal disaster?

o The following text has been added to define a federally-declared disaster and to clarify the designation of eligibility for public assistance under a federal disaster declaration: “Under the Stafford Act, state governors and tribal executives can request federal aid to address materiel, recovery, and economic needs, among others, that surpass local, state, or tribal capacity to address these needs (25). Upon an event being declared a major disaster or emergency by the President of the United States, public assistance grants are disseminated through the U.S. Federal Emergency Management Agency (FEMA) to aid in community and public agency response and recovery efforts (26).”

• Table IV formatting makes it very difficult to read

o The table orientation has been flipped to improve readability.

• IRRs were calculated, but this implies that incidence rates were calculated, but the methods only talk about a population denominator offset . Does this mean cases per person-time was not calculated? If calculated, please add as appendix.

o Rates have been submitted as an appendix.

Discussion:

• Would have appreciated more in-depth discussion of the methodological choices of this study and put them in context with other existing studies (?) assessing a temporal association between disasters and any potential longer-term effects.

o We have added more explicit background on the lack of standard timeframe considered in studies of mental health impacts of disasters (see lines 80 – 103 under Mental health and disasters in the Background).

• The limitations section is too short and not correctly addressing all potential biases or confounding.

o Both the discussion and limitations section have been expanded to address this.

Reviewer #2: This paper provides an empirical analysis examining the impact of natural disasters on violent deaths across five diverse states, each experiencing distinct natural disaster events. Using county-level data, the paper compares violent death counts—including suicides and non-suicide violent deaths—before and after disaster events in both disaster-affected and unaffected counties. The authors adopt a descriptive analytic approach, contrasting patterns of violent deaths pre- and post-disaster across these groups. However, the robustness and clarity of the findings could be significantly enhanced by explicitly employing a rigorous quasi-experimental design, such as a difference-in-differences (DiD) or event-study framework. Baker et al. (2022) and Obradovich et al. (2018) provide valuable examples of how to implement these methodologies effectively in related contexts.

• We appreciate the reviewer noting the value and added rigor of quasi-experimental approaches for assessing changes in a population outcome from before to after a disaster event. We started with a staggered difference-in-differences approach, but we ultimately had to pivot to a simpler analytical approach to better match the quality, completeness, and sample size of our data.

In addition, the discussion of results (e.g., increased incidence of violent deaths post-disaster in affected versus unaffected areas) would benefit from greater consideration of potential spillover effects between counties, a critical factor given the complexity of post-disaster community dynamics.

• We thank the reviewer for bringing up this important topic (see response below). We have not used the term ‘spillover’ because this term has a very different meaning in public health and epidemiology (e.g., when a novel host population comes in contact with a population with a high prevalence of a given pathogen; often in terms of emerging infectious diseases).

Moreover, several aspects of the methodological and analytical choices require further clarity and justification—for instance, the rationale for selecting a three-month period as representative of short-term mental health impacts, and the specific inclusion and treatment of age as a continuous covariate in regression analyses. The manuscript would benefit from improved precision and clarity in reporting statistical methods and results. Specifically, authors should clarify the statistical tests used, implement appropriate adjustments for multiple testing, and explicitly define their regression modeling approach. These refinements would substantially strengthen the manuscript's rigor and interpretability.

Regarding multiple testing, we respectfully disagree. We are testing one a priori stated hypothesis in our analysis – that is comparing the differences in violent death rates for counties that have experienced a disaster from pre- to post-disaster by eligibility for public assistance. The only additional analysis included was stratification by state to account for the effect modification due to characteristics of the state and/or the specific hazard/hazard type. The methods section has been edited to improve clarity. For responses related to the choice of 3-month period, additional details regarding multiple testing adjustments, and treatment of age as a continuous variable, please refer to additional responses below.

Major comments:

1. Improve the statistical design of this study by employing a quasi-experimental design, such as difference-in-difference or event-study frameworks.

Here is a helpful background resource:

Baker, A. C., Larcker, D. F., & Wang, C. C. (2022). How much should we trust staggered difference-in-differences estimates? Journal of Financial Economics, 144(2), 370-395.

Here is a study that applies this methodology to look at the impacts of Katrina on mental health in affected and unaffected counties:

Obradovich, N., Migliorini, R., Paulus, M. P., & Rahwan, I. (2018). Empirical evidence of mental health risks posed by climate change. Proceedings of the National Academy of Sciences, 115(43), 10953-10958.

We thank the reviewer for this suggestion. Our initial model utilized staggered difference-in-difference estimation, but the underlying data were not conducive to model convergence. Therefore, we sought to address our study question using a more straightforward/simple statistical approach which was better suited to the limitations of the underlying data. In future studies, we hope to have more complete and robust data to use the planned staggered DiD estimation. However, the methods section and results have been updated following updated analysis in response to reviewers.

2. Page: 18 lines 220-224: Improve robustness of the discussion of the increased incidence of violent deaths post-disaster in affected and unaffected areas. Please also discuss spillover effects across counties post-disaster.

We have added a discussion of the potential for the spillover of negative social and economic ramifications of disasters for those not eligible for federal public assistance in the third paragraph of the discussion. As noted above, we chose to use an alternative to “spillover” given its different meaning in economics versus epidemiology.

Minor comments:

• Page: 11 lines 76-79: This passage is difficult to follow. Please expand on the community impacts in the context of disasters. It is also unclear what the Australian experience is contradicting to.

o This paragraph has been edited for clarity and to provide more background on the temporal considerations for mental health impacts following disasters.

• Page: 12 line 107: Expand this discussion to extreme weather exposure and violence literature:

Ranson, M., 2014. Crime, weather, and climate change. J. Environ. Econ. Manag. 67, 274–302. https://doi.org/10.1016/j.jeem.2013.11.008

Hsiang, S., Kopp, R., Jina, A., Rising, J., Delgado, M., Mohan, S., Rasmussen, D.J., Muir-Wood, R., Wilson, P., Oppenheimer, M., Larsen, K., and Houser T. (2017). Estimating economic damage from climate change in the United States, Science, 356, 1362–1369

o Additional text has been added to address this concern, reading: “Increasing economic inequality, driven in part by more frequent and severe disasters resulting from climate-related hazards can further exacerbate the negative mental health impacts these events.”

o Additional text has been added to address this concern, reading: “While the relationship between natural hazard exposure and death by homicide or suicide is not well established, it is clear that natural hazard exposure increases perpetration of intimate partner violence and family violence (20) and extreme heat exposure increases a the incidence of various violent behaviors (21).”

• Page: 14 lines 135-137: Please support the specific selection of three months to represent short-term mental health effects. Are the results sensitive to the specific choice of the timeframe?

o Additional references/background has been added to the Mental health and disasters section of the Background to better illustrate the wide range of timelines considered throughout the literature.

o We selected the 3-month time frame in line with the findings of Pietrzak et al., 2012. Added text reads: “Pietrzak et al., (2012), assessed mental health disorders following Hurricane Ike in Galveston and Chambers counties, Texas, at four time points, and found that post-traumatic stress was most prevalent between 2- and 5-months following the event.”

o We have stated more explicitly that findings are not generalizable to other time frames, and future work should investigate violent deaths at different temporal intervals.

• Page: 15 lines 151-152: Earlier you note that age was treated as a continuous variable. How was it introduced in the Poisson regression of aggregate counts in a county/time period?

o Age was not introduced into the regression model; it is only included in the descriptive table to show the underlying characteristics of all deaths.

o Age was treated as a continuous rather than categorical variable for descriptive statistics due to the suppression requirements from CDC that would have rendered reported age categories uninformative.

• Page: 15 lines 154-156: Provide additional information on the regression analysis performed. What was the unit of observation? What controls, beyond affected status and pre/ post time period were included?

o Subsections and explicit definitions have been added under the Analysis section of Methods and Materials.

• Page: 16 lines 178-181: Please elaborate on what the differences were and also in which period (pre/post) these differences were noted.

o This has been updated to reflect the results.

• Page: 16 lines 193-194: Please eliminate discussion of results that did not show up as significant patterns. If a result is not statistically significant, we cannot make claims about protectiveness. Given these wide confidence intervals, it just as easily found be null or >1.

o This has been updated to reflect the results; additionally, both the methods section and results have been updated following updated analysis in response to reviewers.

• Page: 26 Table I: Report the total number of counties in the state.

o Total number of counties for each state has been added to Table I.

• Page: 27 Table II: Add crude incidence rates to this table.

o This table simply shows counts; incidence rates are shown in Table IV. In response to reviewer 1, we have included crude rates in an appendix.

• Page: 28 Table III: Improve labeling. Which test is Chi-Square, and which test is Kruskal-Wallis? Also, apply controls for the multiple testing to the results in this table.

o The “*Kruskal-Wallis Chi-Square p-value” corresponds with the p-values “0.521*” “0.882*). These are the only Kruskal-Wallis p-values in the table, used to test differences in medians between those with and without public assistance. Additional clarification has been added to Table III.

o The use of chi-square p-values for categorical variables and the Kruskal-Wallis chi-square p-value for median age is now clarified in the text, reading: “To compare deaths among those eligible for public assistance and those not eligible in pre-disaster and disaster periods, chi-square p-values are reported for categorical variables (i.e., race, sex, marital status, and education), and Kruskal-Wallis chi-square p-values are reported for median age and interquartile range.”

o Table III only shows descriptive, not inferential, statistics, so there are no controls for multiple testing to be applied. We have changed the title of the table to better reflect the results shown: “Table III. Descriptive statistics of sociodemographic characteristics of all deaths.”

• Page: 29 Table IV: Apply adjustments for multiple testing in this table.

o We thank the reviewer for this comment. We have presented stratified estimates in-line with the stated a-priori research questions: “The aims of this work are 1) to characterize violent deaths in non-disaster and disaster periods and 2) to assess changes in incidence of violent deaths for counties eligible and ineligible for public assistance in non-disaster and disaster periods.”

---

## [Decision Letter · Decision Letter 1]

11 Nov 2025

Dear Dr. Scales,

Thank you for submitting your manuscript to PLOS ONE. After careful consideration, we feel that it has merit but does not fully meet PLOS ONE’s publication criteria as it currently stands. Therefore, we invite you to submit a revised version of the manuscript that addresses the points raised during the review process.

We look forward to receiving your revised manuscript.

Kind regards,

Carolyn Chisadza

Academic Editor

PLOS ONE

Journal Requirements:

Additional Editor Comments:

The revised manuscript has improved substantially in content and quality of analysis. However, there are a few issues mentioned in the comments below that I believe would strengthen the quality of the manuscript. Specifically, the comments related to the Introduction, Methods and Discussion. While the author/s have included the CIs with the IRRs, they can review the comments under the Results as well and address them appropriately.

Reviewers' comments:

**Comments to the Author**

Reviewer #1: All comments have been addressed

Reviewer #2: All comments have been addressed

Reviewer #3: (No Response)

2. Is the manuscript technically sound, and do the data support the conclusions?

Reviewer #1: Yes

Reviewer #2: Yes

Reviewer #3: Yes

3. Has the statistical analysis been performed appropriately and rigorously?

Reviewer #1: Yes

Reviewer #2: Yes

Reviewer #3: Yes

4. Have the authors made all data underlying the findings in their manuscript fully available?

Reviewer #1: Yes

Reviewer #2: No

Reviewer #3: Yes

5. Is the manuscript presented in an intelligible fashion and written in standard English?

Reviewer #1: Yes

Reviewer #2: Yes

Reviewer #3: Yes

Reviewer #1: No further comments from my side, thank you for the opportunity to review this paper. great job and congrats

Reviewer #2: The authors addressed my comments and improved the overall clarity of the paper. I especially appreciated the author's clarifications of their statistical approach choices in their response to the comments document.

Reviewer #3: This manuscript examines whether violent deaths change in the three months after federally declared disasters across five U.S. states, using NVDRS data and FEMA public assistance eligibility as the exposure proxy. The overall framing and use of injury date for event timing are appropriate for an acute-effects question. Several methodological clarifications would strengthen rigor and interpretability.

Abstract

1. Explicitly name the design as an ecological, quasi-experimental pre/post comparison with internal controls and state fixed effects. This prevents readers from over-inferring causal claims.

2. Specify the geographic scope and hazard types in a single sentence to anchor external validity.

3. Report the main effect sizes with 95% confidence intervals for affected and unaffected areas. Briefly note any state-level heterogeneity so readers understand that pooled nulls can mask contrasts.

Introduction

1. Tighten the rationale for the three-month window by linking it to hypothesized acute stress pathways and distinguishing it from longer recovery windows that may show different effects.

2. Justify the choice of FEMA public assistance eligibility as the exposure surrogate. Briefly discuss its relationship to hazard intensity and potential misclassification.

3. Situate the work within prior disaster mortality literature that has focused mainly on suicide. Explain why including non-suicide violent deaths adds value and what you expect to see a priori.

Methods

1. Design and unit of analysis. State plainly that this is an ecological pre/post design with within-state internal controls. Clarify the analysis unit used in models (aggregated eligible vs ineligible counties within state, or county-level panels).

2. Time windows and dating. Define how the three-month pre and post periods are constructed relative to the disaster start date. Confirm that assignment uses injury date rather than death date.

3. Exposure definition. Describe whether FEMA eligibility is county-wide or jurisdiction-specific within counties. Acknowledge cross-boundary injuries and consider a sensitivity analysis assigning exposure by injury county as well as residence.

4. Outcome mapping. List the mapping from NVDRS intents to your three outcomes: all violent deaths, suicide, and other violent deaths. Make it reproducible enough to be replicated.

5. Model specification.

* State the exact model formula, link, offset, and software. Clarify whether the offset reflects population or population-time for the three-month window.

* Report checks for overdispersion and whether you considered negative binomial models or robust standard errors.

* Specify how standard errors were clustered.

6. Covariates and seasonality. Explain why demographic variables are used descriptively rather than in models, given the ecological unit. Address seasonality either by month-matching pre/post windows, including calendar-month fixed effects, or using a county-month panel as a sensitivity analysis.

7. Sample selection. Provide a table listing events per state, the number of affected and unaffected counties, and the population denominators used in each period.

8. Missingness. Report missingness for all descriptive variables. Even if not modeled, patterns of missing data can inform interpretation of group differences.

Results

1. Present crude rates per 100,000 with 95% confidence intervals alongside IRRs for the main contrasts. This grounds effect sizes in absolute terms.

2. State clearly whether overall IRRs differ between pre and post periods for affected and unaffected groups, and do the same for suicides versus other violent deaths. Quote the IRRs and CIs to prevent over-interpretation.

3. Summarize heterogeneity with care. Highlight where non-suicide violent deaths or suicides deviate from the pooled pattern, and pair these statements with uncertainty intervals.

4. Where cell suppression prevents detailed tabulation, consider a forest plot of state-specific IRRs with confidence intervals for each outcome group to communicate patterns without disclosure risk.

Discussion

1. Calibrate language to the study design. Replace causal terms with associational phrasing. Emphasize that the ecological, pre/post design cannot establish causality.

2. Expand limitations on exposure misclassification from using FEMA eligibility, cross-boundary injuries, seasonality, and the loss of information from aggregating counties into affected and unaffected groups. Discuss likely directions of bias.

3. Interpret pooled nulls in light of heterogeneous state-specific results and the focus on acute, largely non-catastrophic events. Clarify that findings may not generalize to catastrophic disasters or to longer recovery periods.

4. Outline next steps: a county-month panel with county and month fixed effects, event-study graphs to check pre-trends, exposure sensitivity using injury county, and alternative count models if overdispersion is present.

Overall, the study is well-motivated and pragmatically designed for an acute-effects question. Clearer design labeling, fuller model diagnostics, explicit handling of seasonality, and results visualizations that foreground uncertainty would materially strengthen the manuscript’s rigor and usefulness.

**Do you want your identity to be public for this peer review?** For information about this choice, including consent withdrawal, please see our Privacy Policy

Reviewer #1: No

Reviewer #2: No

Reviewer #3: **Yes: ** SEPEHR KHOSRAVI

---

## [Author Response · Author response to Decision Letter 2]

12 Nov 2025

Please see attached file for formatted Response to Reviewers. The same content is copied below.

We thank Reviewers 1 and 2 for their substantive feedback and Reviewer 3 for their new feedback. We have addressed the comments and concerns from Reviewer 3, detailed below.

Reviewer #1: No further comments from my side, thank you for the opportunity to review this paper. great job and congrats

Reviewer #2: The authors addressed my comments and improved the overall clarity of the paper. I especially appreciated the author's clarifications of their statistical approach choices in their response to the comments document.

Reviewer #3: This manuscript examines whether violent deaths change in the three months after federally declared disasters across five U.S. states, using NVDRS data and FEMA public assistance eligibility as the exposure proxy. The overall framing and use of injury date for event timing are appropriate for an acute-effects question. Several methodological clarifications would strengthen rigor and interpretability.

Abstract

1. Explicitly name the design as an ecological, quasi-experimental pre/post comparison with internal controls and state fixed effects. This prevents readers from over-inferring causal claims.

• This is now included in the Methods section of the Abstract. This section now reads: “This study uses an ecological, quasi-experimental design to assess violent deaths in pre-disaster and disaster periods across five U.S. states (e.g., North Carolina, Oregon, Oklahoma, Wisconsin, and Colorado) with federally declared disasters resulting from natural hazards (e.g., flood, wildfire, tropical cyclone, storms). Deaths recorded in the National Violent Death Reporting System were used to describe violent deaths with injuries occurring three-months prior to and after disaster onset. Poisson regression with population offsets and fixed effects was used to calculate incidence rate ratios for disaster affected and unaffected counties within the same state, comparing violent death rates in disaster periods with non-disaster periods.”

2. Specify the geographic scope and hazard types in a single sentence to anchor external validity.

• Please see above response to comment 1). The Methods section of the Abstract now reads: “ This study uses an ecological, quasi-experimental design to assess violent deaths in pre-disaster and disaster periods across five U.S. states (e.g., North Carolina, Oregon, Oklahoma, Wisconsin, and Colorado) with federally declared disasters resulting from natural hazards (e.g., flood, wildfire, tropical cyclone, storms).”

3. Report the main effect sizes with 95% confidence intervals for affected and unaffected areas. Briefly note any state-level heterogeneity so readers understand that pooled nulls can mask contrasts.

• Additional clarification has been added in the Results: “Rate ratios were heterogenous but not significant across individual disasters, with only non-suicide and all-cause violent deaths increasing significantly following severe storms and flooding in Wisconsin for counties ineligible for public assistance.”

Introduction

1. Tighten the rationale for the three-month window by linking it to hypothesized acute stress pathways and distinguishing it from longer recovery windows that may show different effects.

• This has been addressed in the Study description and population section of the Materials and methods, with an additional sentence reading: “The three-month disaster period was selected to measure acute disaster-related mental health impacts rather than more longitudinal impacts outside of the immediate response period.”

2. Justify the choice of FEMA public assistance eligibility as the exposure surrogate. Briefly discuss its relationship to hazard intensity and potential misclassification.

a. We added a sentence describing the rationale for using FEMA public assistance eligibility as a proxy for disaster exposure, which now reads: “FEMA public assistance eligibility was used as a proxy measure for disaster exposure because federal assistance eligibility requires a community to have loss and damage that surpasses local and state-level capacity to address resulting acute needs, response, and recovery. It is important to note, however, that smaller scale disasters that are not granted federal assistance can still have tremendous effects on impacted communities, and, even within those eligible for public assistance, there is appreciable variation in need.”

3. Situate the work within prior disaster mortality literature that has focused mainly on suicide. Explain why including non-suicide violent deaths adds value and what you expect to see a priori.

• We have added a sentence to the final paragraph of the Introduction, reading: “By assessing deaths resulting both from suicides and interpersonal violence, this study adds to the literature base for understanding sequelae of violence in disaster contexts.”

• We have added an additional phrase in the final paragraph of the Introduction, which now reads: “Given these complexities, it is critically important to consider the mental health impacts of disasters, including not only violent behaviors but also violent deaths.”

Methods

1. Design and unit of analysis. State plainly that this is an ecological pre/post design with within-state internal controls. Clarify the analysis unit used in models (aggregated eligible vs ineligible counties within state, or county-level panels).

• Additional clarification has been added in the Model specification for incidence rate ratios section of the Materials and methods, reading: “This study uses an ecological pre-post (i.e., pre-disaster and disaster period) design with within-state internal controls (i.e., affected an unaffected counties).”

2. Time windows and dating. Define how the three-month pre and post periods are constructed relative to the disaster start date. Confirm that assignment uses injury date rather than death date.

• This is stated in the sentence including the phrase, “injury data (rather than death date),” but an additional sentence has been added to further clarify this rationale: “Using injury date ensures that the causative injury event preceding a violent death occurred in the temporal range of interest.”

• Three-month pre and post periods are now shown in Table 1 to clarify how these periods were constructed. An additional sentence has been added to read, “The first date of the disaster period is the date of event onset.”

3. Exposure definition. Describe whether FEMA eligibility is county-wide or jurisdiction-specific within counties. Acknowledge cross-boundary injuries and consider a sensitivity analysis assigning exposure by injury county as well as residence.

• This is discussed in the Limitations section: “We used eligibility for federal public assistance to determine if a county was disaster affected. However, the effect of a disaster on a community is considerably more nuanced than the presence or absence of a federal disaster declaration. For example, populations within a community could still experience highly disparate disaster impacts from one another, underscoring the importance of capturing disaster exposure on a more granular scale.”

4. Outcome mapping. List the mapping from NVDRS intents to your three outcomes: all violent deaths, suicide, and other violent deaths. Make it reproducible enough to be replicated.

• The following text has been added in the Outcomes section: “ICD-10 codes corresponding to intentional self-harm or sequelae of intentional self-harm were used to determine suicide deaths. All other deaths were considered ‘non-suicide violent deaths.’ All violent deaths include any death recorded in the NVDRS database, including both suicide and other forms of violent deaths.”

5. Model specification. State the exact model formula, link, offset, and software. Clarify whether the offset reflects population or population-time for the three-month window. Report checks for overdispersion and whether you considered negative binomial models or robust standard errors. Specify how standard errors were clustered.

• These specifications are found in the Analysis and Model specification for incidence rate ratios subsections of the Methods. An additional sentence has been added to the Model specification for incidence rate ratios section, which reads: “Overdispersion was assessed by comparing the mean and variance of the count data, and Poisson regression was an appropriate modelling selection.”

6. Covariates and seasonality. Explain why demographic variables are used descriptively rather than in models, given the ecological unit. Address seasonality either by month-matching pre/post windows, including calendar-month fixed effects, or using a county-month panel as a sensitivity analysis.

• Given small sample sizes, it was not possible to stratify by race, sex, marital status, or education level. We have clarified by adding the sentence, " Because the unit of analysis for this study was the aggregate of affected and unaffected counties, sociodemographic characteristics are reported to contextualize the underlying population but are not included in regression models,” in the Measures section of Materials and Methods.

7. Sample selection. Provide a table listing events per state, the number of affected and unaffected counties, and the population denominators used in each period.

• Please reference Table 1.

8. Missingness. Report missingness for all descriptive variables. Even if not modeled, patterns of missing data can inform interpretation of group differences.

• There was no appreciable missingness for descriptive variables, with missing variables for education shown in Table 3.

Results

1. Present crude rates per 100,000 with 95% confidence intervals alongside IRRs for the main contrasts. This grounds effect sizes in absolute terms.

• These crude rates are presented in supplemental files.

2. State clearly whether overall IRRs differ between pre and post periods for affected and unaffected groups, and do the same for suicides versus other violent deaths. Quote the IRRs and CIs to prevent over-interpretation.

• IRRs are stated in the Results section paragraph following the in-text place holder for Table 3.

3. Summarize heterogeneity with care. Highlight where non-suicide violent deaths or suicides deviate from the pooled pattern, and pair these statements with uncertainty intervals.

• All findings are null, although direction from the null differs. The only deviation from this pattern is described in the Results paragraph following the in-text marker for Table 3, reading: “Only Wisconsin saw a significant increase in non-suicide violent deaths for unaffected counties (IRR: 2.522; 95% CI: 1.600 – 3.979), which drove the observed increased rate of all violent deaths in the disaster period for unaffected counties in Wisconsin.”

4. Where cell suppression prevents detailed tabulation, consider a forest plot of state-specific IRRs with confidence intervals for each outcome group to communicate patterns without disclosure risk.

• We appreciate the Reviewer’s suggestion and, after creating forest plots, have selected to continue with tables to most effectively show results. At the discretion of the editor, forest plots can be submitted as Supplemental files.

Discussion

1. Calibrate language to the study design. Replace causal terms with associational phrasing. Emphasize that the ecological, pre/post design cannot establish causality.

• We have added explicit language around study design and limitations throughout the entirety of the manuscript to ensure that readers do not interpret any results to be causal in nature.

2. Expand limitations on exposure misclassification from using FEMA eligibility, cross-boundary injuries, seasonality, and the loss of information from aggregating counties into affected and unaffected groups. Discuss likely directions of bias.

• Please find this concern addressed in the Limitations section, where we discuss the limitations and potential biases resulting from the selected timeframe, county-level aggregation, cross-border effects, and the use of FEMA public assistance eligibility for disaster exposure.

3. Interpret pooled nulls in light of heterogeneous state-specific results and the focus on acute, largely non-catastrophic events. Clarify that findings may not generalize to catastrophic disasters or to longer recovery periods.

• We have added text noting that catastrophic (e.g., high direct mortality) events could result in different observed violent death rates, given that witnessing deaths is an indicator of negative mental health outcomes, particularly post-traumatic stress disorder.

4. Outline next steps: a county-month panel with county and month fixed effects, event-study graphs to check pre-trends, exposure sensitivity using injury county, and alternative count models if overdispersion is present.

• We have specified several next steps, including studies accounting for the role of displacement and sheltering in the relationship between disasters, stress, and violent deaths and studies evaluating the utilization and effectiveness of mental health programs in disaster affected communities. We also outline the potential biases and limitations resulting from the use of aggregate units of analysis. Finally, we state that future studies should aim to assess the causative relationship of disasters on suicide and violent death mortality rates.

5. Overall, the study is well-motivated and pragmatically designed for an acute-effects question. Clearer design labeling, fuller model diagnostics, explicit handling of seasonality, and results visualizations that foreground uncertainty would materially strengthen the manuscript’s rigor and usefulness.

• We thank the reviewer for their substantive comments, and we believe we have adequately addressed concerns, thereby strengthening the manuscript.

---

## [Editor Report · Decision Letter 2]

16 Nov 2025

Violent deaths following disasters: A retrospective analysis

PONE-D-25-30884R2

Dear Dr. Scales,

We’re pleased to inform you that your manuscript has been judged scientifically suitable for publication and will be formally accepted for publication once it meets all outstanding technical requirements.

Kind regards,

Carolyn Chisadza

Academic Editor

PLOS ONE
---

## [Editor Report · Acceptance letter]

PONE-D-25-30884R2

PLOS ONE

Dear Dr. Scales,

I'm pleased to inform you that your manuscript has been deemed suitable for publication in PLOS ONE. Congratulations! Your manuscript is now being handed over to our production team.

Kind regards,

on behalf of

Prof Carolyn Chisadza

Academic Editor

PLOS ONE